# Association of Dual Sensory Impairment with Declining Physical Function in Community-Dwelling Older Adults

**DOI:** 10.3390/ijerph20043546

**Published:** 2023-02-17

**Authors:** Hyun Ho Kong, Kwangsoo Shin, Chang Won Won

**Affiliations:** 1Department of Rehabilitation Medicine, Chungbuk National University Hospital, Cheongju 28644, Republic of Korea; 2Graduate School of Public Health and Healthcare Management, Songeui Medical Campus, The Catholic University of Korea, Seoul 06591, Republic of Korea; 3Elderly Frailty Research Center, Department of Family Medicine, College of Medicine, Kyung Hee University, Seoul 02447, Republic of Korea

**Keywords:** sensation disorders, hearing loss, vision disorders, muscle strength, physical functional performance

## Abstract

Few studies have investigated whether dual sensory impairment (DSI) adversely affects the deterioration of physical function in older adults compared to single sensory impairment (SSI, visual or auditory). We studied the association between DSI and declining physical function by analyzing the data of 2780 Korean community-dwelling adults aged 70–84 years. Sensory impairment was assessed through pure tone audiometry and visual acuity testing. Muscle strength (handgrip strength) and physical performance (timed up and go test and short physical performance battery (SPPB)) were evaluated. In the cross-sectional analysis, DSI was associated with higher odds of having low muscle strength (odds ratio (OR), 1.78; 95% confidence interval (CI), 1.27–2.48) and poor physical performance (SPPB: OR, 2.04; 95% CI, 1.38–3.00) than SSI. Among all sensory impairment groups in the longitudinal analysis, DSI at baseline increased the risk of deteriorating physical performance during the follow-up period (OR, 1.94; 95% CI, 1.31–2.88; *p* < 0.01) the most. DSI showed a more severe adverse effect on the decline in physical function among community-dwelling older adults than SSI. More comprehensive care is needed to prevent the deterioration of physical function in older adults due to DSI.

## 1. Introduction

The incidence rate of sensory impairments (vision, hearing, smell, taste, and touch) increases with age [1]. Particularly, visual impairment (VI) and hearing impairment (HI) are common in older adults due to age-related neurophysiologic changes in vision (spatial contrast sensitivity, density of photoreceptors, visual processing speed, etc.) and hearing (degeneration of hair cells within the cochlea, atrophy of the spiral ganglion cells, diminished grey matter volume in the primary auditory cortex, etc.) [2,3]. It has been reported that 86% (31 million) of blind people (36 million) and 80% (172.3 million) of persons with moderate to severe VI (216.6 million) worldwide are older than 50 years [4]. In addition, about two-thirds of the population older than 70 years suffer from HI in the United States [5], and it is expected that 585 million people older than 65 years worldwide will experience HI by 2050 [6].

Sensory impairment, in the form of HI and VI, is associated with several problems experienced by older adults. Firstly, it is associated with a decline in physical characteristics, such as muscle strength, gait, and cardiorespiratory fitness [7,8,9], and deteriorating physical function increases the risk of falls and fractures in older adults [10,11,12]. In addition, sensory impairment is related to limitations in performing activities of daily living (ADLs) [13], reduced social interaction [14], and deteriorating quality of life in older adults [15]. It has also been reported to increase the risk of cognitive decline and dementia [16].

In particular, dual sensory impairment (DSI), a condition in which VI and HI coexist, can be expected to result in more severe adverse effects in terms of various forms of functional decline in older adults compared with single sensory impairment (SSI). Previous studies have reported that older adults with DSI find it more difficult to engage in ADLs [17] and have a higher risk of depression [18] and cognitive decline than those with SSI [16,18].

In addition, DSI is known to result in more severe adverse effects, in terms of deterioration of physical function, than SSI in older adults [7,19]. However, a majority of studies about the effects of DSI on physical function in older adults were limited in that they evaluated sensory impairment and physical function using self-report questionnaires rather than objective and quantitative methods [7,19,20]. Furthermore, it was difficult to elucidate the causal relationship between DSI and the decline in physical function since most of these studies had a cross-sectional design [7,20,21].

The purpose of the present study was to investigate the effect of DSI on physical function among community-dwelling older adults by analyzing large-scale cohort data derived from participants subjected to sensory impairment evaluation via objective methods (pure tone audiometry (PTA) and visual acuity testing) and assessments of physical function (including handgrip strength and the short physical performance battery (SPPB)) by trained investigators.

## 2. Materials and Methods

### 2.1. Study Population

This study was conducted utilizing the data of the Korean Frailty and Aging Cohort Study, a longitudinal multicenter cohort study conducted between 2016 and 2019 to identify the causes of frailty in community-dwelling older persons (aged 70–84 years) and how to prevent it. In the baseline survey (2016–2017), the participants were recruited stratified by age and sex from ten urban and rural areas nationwide in Korea, and each site accounted for approximately 300 participants. Each participant underwent an individual interview, health examination, and physical function tests every 2 years [22].

A total of 3014 participants were recruited in the baseline study conducted between 2016 and 2017. Eighty-five of these participants without data on physical function and 41 participants without data on the assessment of VI or HI were excluded from the analysis. Participants without information on the following variables that would be used as covariates were also excluded from the analysis: education level (n = 2), alcohol consumption history (n = 5), smoking history (n = 18), and underlying diseases (n = 83). Finally, 2780 patients (1325 men and 1455 women) were included in the baseline cross-sectional analysis.

A longitudinal analysis was performed on 2346 participants, excluding 434 participants who did not undergo follow-up physical function tests due to refusal for investigation or death between 2018 and 2019, 2 years after the baseline study.

### 2.2. Assessment of Sensory Impairment

Hearing acuity was measured via PTA (with four frequencies: 0.5, 1.0, 2.0, and 4.0 kHz) and participants using hearing aids were allowed to use them during evaluation. The pure tone average was defined as the average of hearing thresholds at each frequency. HI was defined as a pure tone average of the better-hearing ear of greater than 40 dB with reference to previous studies [23,24,25].

Visual acuity was measured using Dr. Hahn’s standard vision chart, which is based on the Snellen chart at a distance of 4 m. Participants using corrective lenses were assessed with them on. VI was defined in participants among whom the best corrected visual acuity of the better-seeing eye was worse than 20/40 [26].

### 2.3. Assessment of Physical Function

Physical function in older adults was classified into muscle strength (handgrip strength) and physical performance (timed up and go (TUG) test and SPPB) [27]. Handgrip strength was measured twice in both hands at 3-minute intervals using a digital grip dynamometer (Takei TKK 5401; Takei Scientific Instruments, Tokyo, Japan), and the highest recorded value was used for analysis. The TUG test was performed by measuring the time it takes to get up from a chair without an armrest, walk 3 m, turn around, and return to the chair to sit down [28]. The SPPB had three components: balance, gait speed, and a five-repetition chair stand test (CS-5). Each component was scored on a scale of 0 to 4, with the total score ranging from 0 to 12 points. The balance test consisted of side-by-side, semi-tandem and tandem stands, and scores were allocated while standing for 10 s in each posture. The CS-5 measures the time taken to stand five times from a sitting position from a straight-backed armchair without using the arms. The gait speed was defined as the fastest time in two trials required for participants to walk 4 m (with acceleration and deceleration phases of 1.5 m each) at the usual gait speed [29]. Low muscle strength was defined as a handgrip strength < 28 kg in men and <18 kg in women. Poor physical performance was defined as a composite SPPB score ≤ 9 [27].

### 2.4. Adjustment Covariates

Age, sex, body mass index (BMI), education level (below elementary school, elementary school, middle or high school, or beyond high school), alcohol consumption, and smoking history were investigated as demographic factors. In addition, underlying diseases (including cerebrovascular diseases, cardiac diseases, chronic obstructive pulmonary disease, asthma, diabetes mellitus, and arthritis) that could affect the analysis as covariates were investigated via a self-administered questionnaire.

### 2.5. Statistical Analysis

To compare baseline demographic characteristics and physical function between groups according to sensory impairment, analysis of variance and chi-square tests were performed for continuous and categorical variables, respectively. Continuous variables are expressed as means and standard deviations and categorical variables are expressed as numbers and percentages.

Generalized linear modeling (GLM) was performed to investigate the association between sensory impairment and physical function, with adjustment for confounders identified in the analysis of baseline demographics.

Multivariable logistic regression analysis (as a baseline cross-sectional study) was performed to investigate the risk of low physical function (low muscle strength or poor physical performance) according to sensory impairment in older adults. In addition, a generalized estimating equation (GEE) model for longitudinal studies was conducted to analyze the baseline effect of sensory impairment on the decline in physical function in older adults during the follow-up period. 

We performed all statistical analyses using SPSS version 25.0 (IBM Corporation, Chicago, IL, USA) and analysis items with *p*-values < 0.05 were considered statistically significant.

## 3. Results

### 3.1. Baseline Characteristics 

A total of 2780 participants were included in the baseline analysis. The mean age of the participants was 75.9 ± 3.9 years, and 47.7% of the participants were males. The normal sensory (NS) group comprised 1501 participants (54.0%), and there were 883 participants (31.8%) in the HI only group, 209 (7.5%) in the VI only group, and 187 (6.7%) in the DSI group.

Regarding demographic characteristics, the participants’ ages in the sensory impairment groups were significantly higher than those in the NS group; in particular, the DSI group had the highest mean age compared with the other groups (*p* < 0.001). Compared with females, males had higher rates of HI and lower rates of DSI (*p* < 0.05). The DSI group had a lower mean BMI than the NS group (*p* < 0.05), and the DSI group had a lower education level than the NS group (*p* < 0.001). Additionally, the DSI group had a higher proportion of ex-smokers or current smokers than the NS group (*p* < 0.01). However, there were no significant differences in alcohol consumption and the number of underlying diseases (including cerebrovascular disease) between the groups according to sensory impairment.

In the comparison of physical function in each group according to sensory impairment, muscle strength (handgrip strength) was statistically significantly lower in the VI only (24.1 ± 7.5 kg) and DSI (22.8 ± 6.9 kg) groups than in the NS group (25.8 ± 7.6 kg) (*p* < 0.001). Regarding physical performance, the DSI group showed the poorest TUG test performance compared with the other groups (NS vs. DSI, 10.0 ± 2.3 s vs. 11.9 ± 3.5 s, *p* < 0.001). Statistically, the DSI group showed the lowest values of gait speed (NS vs. DSI: 1.1 ± 0.2 m/s vs. 1.0 ± 0.3 m/s, *p* < 0.001), CS-5 (NS vs. DSI: 11.1 ± 3.4 s vs. 13.1 ± 5.2 s, *p* < 0.001), and composite SPPB score (NS vs. HI only vs. VI only vs. DSI: 11.0 ± 1.3 vs. 10.8 ± 1.5 vs. 10.7 ± 1.4 vs. 10.1 ± 1.9 points; *p* < 0.001) (Table 1).

### 3.2. Association of Sensory Impairment with Physical Function

In the GLM analysis, HI was associated with poor TUG test performance (β = 0.32; 95% confidence interval (CI), 0.22 to 0.41; *p* < 0.01) and low SPPB scores (β = −0.16; 95% CI, −0.21 to −0.10; *p* < 0.01) (Table 2). VI was significantly associated with low muscle strength (β = −1.18; 95% CI, −1.45 to −0.91; *p* < 0.001) as well as poor TUG test performance (β = 0.52; 95% CI, 0.39 to 0.65; *p* < 0.001) and low SPPB scores (β = −0.26; 95% CI, −0.33 to −0.18; *p* < 0.001).

In the comparison of groups according to sensory impairment (category 3), the VI only (β = −1.24; 95% CI, −1.61 to −0.87; *p* < 0.01) and DSI (β = −1.39; 95% CI, −1.78 to −1.00; *p* < 0.001) groups showed a significant association with lower muscle strength compared with the NS group. In terms of physical performance, the DSI group showed an association with poor TUG test performance (β = 1.08; 95% CI, 0.89 to 1.26; *p* < 0.001) and low SPPB composite scores (β = −0.51; 95% CI, −0.61 to −0.40; *p* < 0.001) compared with the other groups. In addition, the HI only group was significantly associated with poor TUG test performance (β = 0.20; 95% CI, 0.10 to 0.30; *p* < 0.05).

In the sub-analysis of each component test of the SPPB, the HI group showed a significantly poor performance only in the balance test (β = −0.10; 95% CI, −0.13 to −0.08; *p* < 0.001). In further comparisons between groups according to sensory impairment, the DSI group showed a significant correlation with lower physical performance in all sub-tests (balance, CS-5, and gait speed tests) of the SPPB compared with the other groups (*p* < 0.001) (Table 2).

### 3.3. Association of DSI with Low Physical Function (Cross-Sectional Analysis) 

At baseline, there were 770 participants in the low muscle strength group and 471 participants in the poor physical performance group. In the multivariable logistic regression analysis, the odds of low muscle strength were higher in the VI only (odds ratio (OR), 1.51; 95% CI, 1.10 to 2.08; *p* < 0.05) and DSI (OR, 1.78; 95% CI, 1.27 to 2.48; *p* < 0.01) groups than in the NS group. Additionally, compared with the NS group, the HI only (OR, 1.39; 95% CI, 1.10 to 1.77; *p* < 0.01) and VI only (OR 1.72, 95% CI, 1.18 to 2.50, *p* < 0.01) groups showed higher odds of poor physical performance, and the DSI group showed the highest odds (OR, 2.04; 95% CI, 1.38 to 3.00; *p* < 0.001) (Table 3).

### 3.4. Association of DSI with Low Physical Function (Longitudinal Analysis)

Regarding physical function, 250 and 288 participants with normal physical function at baseline developed a low muscle strength (handgrip strength) and physical performance (SPPB), respectively. In the GEE model, VI only (OR, 1.61; 95% CI, 1.14 to 2.28; *p* < 0.01) and DSI (OR, 1.51; 95% CI, 1.04 to 2.18; *p* < 0.05) at baseline were associated with low muscle strength during the follow-up period. In addition, DSI at baseline was associated with the highest odds of declining physical performance (SPPB: OR, 1.94; 95% CI, 1.31 to 2.88; *p* < 0.01) during the follow-up period compared with other groups (Table 4). 

## 4. Discussion

In this study, DSI was associated with a lower physical function in terms of muscle strength (handgrip strength) and poorer physical performance (TUG test and SPPB) than SSI. In addition, the participants with DSI at baseline showed higher odds of declining physical function during the follow-up period than other groups.

In both cross-sectional and longitudinal analyses of this study, sensory impairments increased the risk of deteriorating physical function in older adults. Considering that tiny differences in physical functions (e.g., gait speed: hazard ratio (HR) 0.88 per each 0.1 m/s faster, handgrip strength: HR 0.96 per 1 kg higher) can affect prognosis, such as mortality in older adults [30,31], a statistically significant difference in the physical function between the NS group and the DSI group in this study is considered to be clinically relevant. In a previous study, severe HI was associated with poor physical performance (measured via the SPPB), with a faster physical performance decline compared to the NS group [8]. Furthermore, it has been reported that VI is associated with a slower walking speed and shorter one-leg standing time [9]. Further, compared with SSI, DSI increases the risks of weak handgrip strength [7] and slow walking speed to greater extents among older adults [32]. However, it has been difficult to quantify the effect of DSI on physical function in older adults because most previous studies evaluated sensory impairment and physical function via a self-report method, which could be subjective [7,19,20]. Here, we could minimize the limitations of previous studies in which measurement data were under- or overestimated due to participant bias by objectively quantifying each sensory impairment and physical function.

The mechanisms by which sensory impairment (in the forms of VI and HI) adversely affects physical function in older adults are unclear, although various causes can be assumed. First, HI is associated with deteriorating balance function as a component of physical function in older adults because problems of the auditory system are likely to be accompanied by abnormalities of the vestibular system belonging to the same inner ear organ, and this may cause abnormalities in balance [33]. Previous studies reported that participants with HI had a high tendency to experience decreased balance function [34]. In the present study, the HI group also showed a significant association with deteriorating performance in the TUG test and balance test in the SPPB (Table 2). Second, older adults with VI find it difficult to engage in ADL [35] and have an increased risk of falls [36], resulting in a gradual decrease in physical activity and leading to a decrease in overall physical function. In this study, unlike HI, which showed an adverse effect mainly on balance, VI showed a correlation with deteriorating overall physical function in older adults, including muscle strength (assessed via handgrip strength and CS-5 tests) and physical performance assessed via the TUG test and SPPB. Third, sensory impairment in older adults may be an early marker of deteriorating physical function because sensory impairment and deteriorating physical function in older adults share common causative factors, such as aging, vascular diseases, and inflammation [37,38,39]. Since participants with DSI cannot compensate for unimpaired sensory inputs unlike participants with SSI, it is presumed that they have a higher risk of declining physical function than participants with SSI.

In the GLM analysis of this study, HI was associated with lower physical performance (measured in the TUG test and balance test of the SPPB), but not with muscle strength (measured via handgrip strength and CS-5 tests) in older adults (Table 2). The same results were obtained in the multivariable logistic regression analysis and GEE model (Table 3 and Table 4). Several previous studies have also reported that HI alone did not correlate with declining handgrip strength [40,41]. Considering the results of previous studies and the present study, we can assume that HI mainly affects physical performance (including balance function) rather than muscle strength in a static position. However, considering that DSI was more strongly associated with low muscle strength than VI only (Table 3), it is presumed that auditory function, which is intact in individuals with VI alone, contributes an important role in sensory compensation. 

In the longitudinal analysis, DSI at baseline was associated with the highest increase in the risk of deteriorating physical performance during the follow-up period (OR, 1.94; 95% CI, 1.31 to 2.88; *p* < 0.01) (Table 4). Despite the short follow-up period of 2 years, these results are considered to indicate that DSI results in a greater acceleration of the decline in physical performance in older adults than SSI. In a previous longitudinal study on the effect of sensory impairment on the ability to perform ADLs in older adults, the risk of decline in instrumental ADL ability was significantly higher in participants with DSI than in those with SSI [17]. However, there was no difference in the incidence of low muscle strength during the follow-up period between the participants with DSI and those with VI only at baseline. Considering that the average annualized decline in handgrip strength was approximately 1% [42], we assumed that the follow-up period of this study was too short to verify the significance of the declining muscle strength due to DSI.

The present study had several limitations. First, the longitudinal follow-up period of 2 years was too short to observe the effect of sensory impairment on physical function. As observed in previous studies, DSI may appear to be a high risk factor for the decline in physical function in short-term observations; however, in long-term observations, the effect may gradually diminish over time [43] or vice versa [44]. We expect that a clearer analysis will be possible through an additional longitudinal follow-up study. Second, additional potential confounding factors were not measured in the present study. In particular, participants with HI have an increased likelihood of having an accompanying vestibular disorder, which may have lowered physical function in participants with HI; however, vestibular balance disorders were not assessed in the survey of underlying diseases in this cohort. For this reason, the possibility that confounding occurred in the analysis cannot be excluded. Third, since this cohort only included community-dwelling Asians, it is difficult to generalize the results of this study to people of other origins or older adults receiving care services at residential facilities.

Nevertheless, the present study has several strengths. First, most previous studies investigated sensory impairment and physical function via self-reported questionnaire methods, such that the measurement data may be under- or overestimated due to participant bias; however, the present study was able to minimize the limitations of previous studies by assessing sensory impairment through PTA and visual acuity tests and various quantitative physical function tests. Second, the causal relationship between sensory impairment and the decline in physical function could be confirmed through longitudinal analysis. Third, reliable study results were obtained through the analysis of nationwide large-scale and well-organized cohort data by including participants in urban and rural areas.

## 5. Conclusions

DSI increases the risk of deteriorating physical function (muscle strength and physical performance) to a greater extent than SSI in community-dwelling older adults. Therefore, more attention should be paid to the diagnostic evaluation and management of sensory impairment aimed at slowing the rate of decline in physical function in older adults.

## Figures and Tables

**Table 1 ijerph-20-03546-t001:** Characteristics and physical functions of the participants according to sensory impairment.

Variables	NS	HI Only	VI Only	DSI	Total	*p*-Value
Age, years; mean (SD)	75.1 (3.6)	76.8 (3.9) ^a^	76.2 (3.8) ^a^	78.5 (3.6) ^abc^	75.9 (3.9)	<0.001
Sex, n (%)						<0.05
Males	638 (42.5)	513 (58.1)	94 (45.0)	80 (42.8)	1325 (47.7)	
Females	863 (57.5)	370 (41.9)	115 (55.0)	107 (57.2)	1455 (52.3)	
BMI, kg/m^2^; mean (SD)	24.5 (2.9)	24.4 (3.1)	24.1 (3.2)	23.9 (3.2) ^a^	24.4 (3.0)	<0.05
Education level, n (%)						<0.001
<Elementary	252 (16.8)	161 (18.2)	60 (28.7)	75 (40.1)	548 (19.7)	
Elementary	406 (27.0)	228 (25.8)	67 (32.1)	50 (26.7)	751 (27.0)	
Middle or high school	558 (37.2)	326 (36.9)	61 (29.2)	43 (23.0)	988 (35.5)	
>High school	285 (19.0)	168 (19.0)	21 (10.0)	19 (10.2)	493 (17.7)	
Alcohol consumption, n (%)	1042 (69.4)	645 (73.0)	156 (74.6)	134 (71.7)	1977 (71.1)	0.10
Smoking history, n (%)						<0.01
Never smoked	986 (65.7)	482 (54.6)	132 (63.2)	114 (61.0)	1714 (61.7)	
Ex-smoker	433 (28.8)	350 (39.6)	63 (30.1)	56 (29.9)	902 (32.4)	
Current smoker	82 (5.5)	51 (5.8)	14 (6.7)	17 (9.1)	164 (5.9)	
Number of underlying diseases, n (%)						0.52
0	750 (50.0)	459 (52.0)	100 (47.8)	85 (45.5)	1394 (50.1)	
1	578 (38.5)	304 (34.4)	90 (43.1)	78 (41.7)	1050 (37.8)	
2	148 (9.9)	100 (11.3)	17 (8.1)	22 (11.8)	287 (10.3)	
≥3	25 (1.7)	20 (2.3)	2 (1.0)	2 (1.1)	49 (1.8)	
Cerebrovascular disease (+), (%)	64 (4.3)	44 (5.0)	11 (5.3)	8 (4.3)	127 (4.6)	0.61
Physical function						
Handgrip strength, kg; mean (SD)	25.8 (7.6)	26.5 (7.3)	24.1 (7.5) ^ab^	22.8 (6.9) ^ab^	25.7 (7.5)	<0.001
TUG, s; mean (SD)	10.0 (2.3)	10.5 (2.5) ^a^	10.6 (2.3) ^a^	11.9 (3.5) ^abc^	10.4 (2.5)	<0.001
Gait speed, m/s; mean (SD)	1.1 (0.2)	1.1 (0.3)	1.1 (0.3)	1.0 (0.3) ^abc^	1.1 (0.2)	<0.001
CS-5, s; mean (SD)	11.1 (3.4)	11.5 (4.1)	11.8 (3.7)	13.1 (5.2) ^abc^	11.4 (3.8)	<0.001
Composite SPPB score, points; mean (SD)	11.0 (1.3)	10.8 (1.5) ^a^	10.7 (1.4) ^a^	10.1 (1.9) ^abc^	10.9 (1.4)	<0.001

Note: NS, normal sensory; HI, hearing impairment; VI, visual impairment; DSI, dual sensory impairment; SD, standard deviation; BMI, body mass index; TUG, timed up and go; CS-5, five-repetition chair stand test; SPPB, short physical performance battery. Underlying diseases: cerebrovascular diseases, cardiac diseases, chronic obstructive pulmonary disease, asthma, diabetes mellitus, and arthritis. ^a^ Significant differences among the NS and HI only/VI only/DSI groups. ^b^ Significant differences among the HI only and VI only/DSI groups. ^c^ Significant difference between the VI only and DSI groups.

**Table 2 ijerph-20-03546-t002:** Association of sensory impairment with physical function (cross-sectional analysis).

**Outcome Variables**		**β (95% CI)**	
	**Handgrip Strength**	**TUG**	**SPPB Composite Score**
Category 1			
Normal	Reference	Reference	Reference
HI	−0.31 (−0.51 to −0.11)	0.32 (0.22 to 0.41) ^b^	−0.16 (−0.21 to −0.10) ^b^
Category 2			
Normal	Reference	Reference	Reference
VI	−1.18 (−1.45 to −0.91) ^c^	0.52 (0.39 to 0.65) ^c^	−0.26 (−0.33 to −0.18) ^c^
Category 3			
Normal	Reference	Reference	Reference
HI only	−0.29 (−0.51 to −0.08)	0.20 (0.10 to 0.30) ^a^	−0.11 (−0.17 to −0.05)
VI only	−1.24 (−1.61 to −0.87) ^b^	0.22 (0.05 to 0.39)	−0.12 (−0.22 to −0.02)
DSI	−1.39 (−1.78 to −1.00) ^c^	1.08 (0.89 to 1.26) ^c^	−0.51 (−0.61 to −0.40) ^c^
**Outcome Variables**		**β (95% CI)**	
	**Balance**	**CS-5**	**Gait Speed**
Category 1			
Normal	Reference	Reference	Reference
HI	−0.10 (−0.13 to −0.08) ^c^	0.24 (0.10 to 0.39)	−0.002 (−0.011 to 0.007)
Category 2			
Normal	Reference	Reference	Reference
VI	−0.11 (−0.14 to −0.08) ^c^	0.54 (0.34 to 0.73) ^b^	−0.016 (−0.028 to −0.004)
Category 3			
Normal	Reference	Reference	Reference
HI only	−0.08 (−0.11 to −0.06) ^b^	0.13 (−0.03 to 0.29)	0.004 (−0.006 to 0.014)
VI only	−0.05 (−0.09 to −0.01)	0.26 (0.00 to 0.53)	0.000 (−0.017 to 0.017)
DSI	−0.25 (−0.29 to −0.20) ^c^	1.02 (0.74 to 1.31) ^c^	−0.033 (−0.051 to −0.015) ^c^

Note: ^a^ *p* < 0.05; ^b^ *p* < 0.01; ^c^ *p* < 0.001. The p-value is based on the generalized linear modeling adjusted for age, sex, body mass index, education level, and smoking history.

**Table 3 ijerph-20-03546-t003:** Association between sensory impairment and low physical function (cross-sectional analysis).

	**Low Muscle Strength (Handgrip Strength)**					
	**Unadjusted**		**Model 1**		**Model 2**	
	OR (95% CI)	*p*-value	OR (95% CI)	*p*-value	OR (95% CI)	*p*-value
HI only	1.34 (1.11–1.61)	<0.01	1.07 (0.88–1.32)	0.50	1.04 (0.85–1.27)	0.71
VI only	1.88 (1.38–2.55)	<0.001	1.71 (1.25–2.34)	<0.01	1.51 (1.10–2.08)	<0.05
DSI	2.99 (2.19–4.08)	<0.001	2.05 (1.48–2.84)	<0.001	1.78 (1.27–2.48)	<0.01
	**Poor Physical Performance (SPPB)**					
	**Unadjusted**		**Model 1**		**Model 2**	
	OR (95% CI)	*p*-value	OR (95% CI)	*p*-value	OR (95% CI)	*p*-value
HI only	1.51 (1.21–1.89)	<0.001	1.42 (1.12–1.80)	<0.01	1.39 (1.10–1.77)	<0.01
VI only	2.14 (1.50–3.05)	<0.001	1.94 (1.35–2.80)	<0.001	1.72 (1.18–2.50)	<0.01
DSI	3.21 (2.25–4.59)	<0.001	2.34 (1.61–3.41)	<0.001	2.04 (1.38–3.00)	<0.001

Note: OR, odds ratio; CI, confidence interval. Low muscle strength was defined as handgrip strength < 28 kg in males and <18 kg in females. Poor physical performance was defined as a composite SPPB score ≤ 9. Model 1: adjusted for age and sex; Model 2: adjusted for age, sex, body mass index, education level, and smoking history.

**Table 4 ijerph-20-03546-t004:** Association between sensory impairment and low physical function (longitudinal analysis).

	**Low Muscle Strength (Handgrip Strength)**					
	**Unadjusted**		**Model 1**		**Model 2**	
	OR (95% CI)	*p*-value	OR (95% CI)	*p*-value	OR (95% CI)	*p*-value
HI only	1.50 (1.23–1.84)	<0.001	1.21 (0.98–1.50)	0.08	1.19 (0.96–1.49)	0.11
VI only	2.13 (1.53–2.97)	<0.001	1.82 (1.29–2.57)	<0.01	1.61 (1.14–2.28)	<0.01
DSI	2.62 (1.85–3.71)	<0.001	1.68 (1.17–2.40)	<0.01	1.51 (1.04–2.18)	<0.05
	**Poor Physical performance (SPPB)**					
	**Unadjusted**		**Model 1**		**Model 2**	
	OR (95% CI)	*p*-value	OR (95% CI)	*p*-value	OR (95% CI)	*p*-value
HI only	1.51 (1.21–1.89)	<0.001	1.36 (1.07–1.72)	<0.05	1.34 (1.06–1.71)	<0.05
VI only	2.14 (1.50–3.05)	<0.001	1.87 (1.28–2.72)	<0.01	1.67 (1.14–2.47)	<0.01
DSI	3.21 (2.25–4.59)	<0.001	2.20 (1.51–3.21)	<0.001	1.94 (1.31–2.88)	<0.01

Note: Low muscle strength was defined as handgrip strength < 28 kg in males and <18 kg in females. Poor physical performance was defined as a composite SPPB score ≤ 9. Model 1: adjusted for age and sex; Model 2: adjusted for age, sex, body mass index, education level, and smoking history.

## Data Availability

Data sharing is not applicable to this article.

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
