# Peer review of "Association of Dual Sensory Impairment with Declining Physical Function in Community-Dwelling Older Adults"

_ijerph, 2023, doi:10.3390/ijerph20043546_

Round 1

Reviewer 1 Report

This is a very interesting study that aimed to investigate the association between dual sensory impairment and physical decline. The sample size is representative and the statistical analyses brought interesting results. In that sense, I congratulate the authors. The manuscript, however, needs to be revised and some suggestions and questions are presented here for good application of the findings.

 1) General aspects: How did you select the 2780 participants? Using the term “cohort” means that this is a longitudinal study.  However, the data are presented in a cross-section design. Please clarify this matter in the text. If the authors assessed subjects’ cognitive functions, they should include this factor in the analyses (as cognitive decline is as common as sensory impairment in the elderly).

 2) Title: Do you think that using the term “The Korean Study” in title would make the manuscript less attractive to researchers outside South Korea?

 3) Abstract: Adequate.

 4) Keywords: Use only keywords registered in MESH.

 5) Introduction: Authors should furthrer more this part of the text. There are other types of sensory impairment rather than hearing or visual impairment. I suggest the inclusion of more information about proprioception, stereognosis, tactile anesthesia etc.

 6) Introduction: I suggest that authors use hearing impairment and visual impairment instead of “HI” or “VI” (avoid the abbreviation forms).

 7) Introduction: Add more neurophysiology to explain sensory deficits in the elderly.

 8) Methods: The World Health Organization classify older adults as a person > 60 years. Why did you include subjects > 70 years?

 9) Methods: Cognitive decline is common in elderly. If you have collect any data on cognition, it should be interesting to be included in the analyses.

 10) Methods: Please explain why you used hearing and visual impairment instead of other forms of sensory impairment.

 11) Methods: Are there cases of patients with neurological dysfunctions such as stroke, spinal cord injury, etc.? These conditions cause sensory impairment as well.

 12) Results: Adequate. Congrats!

 13) Discussion: Some analyses showed statistical differences. However, not all are clinically relevant. For example, TUG scores ranged between 10.0 and 11.9 steps. Is this difference clinically relevant? I don’t think so. Authors should discuss more this aspect.

14) Discussion: The same for handgrip strength, gait speed, CS-5, and SPPB. The differences between groups are statistically significant, but not clinically relevant.

15) Conclusion: Do not use references is the conclusion section. This is mandatory.

16) References: Adequate.

Author Response

I appreciate the reviewer's detailed comments. We uploaded detailed responses to the reviewer's comments in the form of a Word file. Please see the attachment. 

Reviewer 2 Report

In this study, the authors investigated whether hearing and/or vision loss in community-dwelling elderly Koreans of both sexes (age 70+) affected their muscle strength and general physical performance. The research is longitudinal and the same parameters were measured two years after the initial research. Compared to other studies that mainly examine the subjective impression of respondents, here they used objective methods.  Although it can be said that all the respondents were old, there were differences between them: the average age of people with hearing or vision loss (or both senses) was significantly higher than the average age of people whose tested senses still functioned well at the beginning of the study. Longitudinally, the subjects of both sexes in whom the loss of both tested senses was detected at the start, showed a greater deviation of bodily functions than persons who at the beginning had the loss of only one sense.

Generally speaking, I only have positive things to say about this research. The study is well positioned, the literature is appropriately reviewed, the data collection efforts are well described, the results are clearly articulated and justified, and limitations are acknowledged. I don’t have any major concerns.

I just have some minor comments/objections:

The authors stated that subjects who normally regularly use hearing and visual aids were allowed to use these aids during testing (lines 86-93). It is not clear whether some of these people were identified as hearing- and/or sight-impaired despite the use of aids? If so, please indicate the proportion of respondents who used visual and/or hearing aids before the survey in the newly-defined groups (defined using objective measurements) of “Visually impaired”, “Hearing impaired” and “Dual sensory impairment”.

page 5, line 170 – please refer to Table 2 at the end of the first sentence (these paragraph is too long to mention it only at the end)

page 5, lines 173-179 - you did not refer to a statistically significant difference in TUG results between normal and HI subjects

page 6, line 207 - since only the first row is visible here, please put the entire table on the next page (so that it can be seen when it is printed)

page 7, line 217 – please change “compared with” to “compared to”

page 8, line 284 – please avoid using the term “other races”, I suggest you change it to “people of other origins”

Author Response

(The authors gave the same response as above.)
